# PRIVACY-PRESERVING LLM FINE-TUNING OVER API

## ABSTRACT

As deep learning models become larger and more expensive, many practitioners turn to fine-tuning APIs. These web services allow fine-tuning a model between two parties: the client that provides the data, and the server that hosts the model. While convenient, the fine-tuning APIs raise a new concern: the data of the client is at risk of privacy breach during the training procedure. This challenge presents an important practical case of vertical federated learning, where the two parties perform parameter-efficient fine-tuning (PEFT) of a large pre-trained model. In this study, we systematically search for a way to fine-tune models over an API *while keeping the labels private*. We analyze the privacy of popular algorithms for parameter-efficient fine-tuning when training over an API. Using this analysis, we propose P³EFT, a multi-party split learning algorithm that takes advantage of existing PEFT properties to maintain privacy at a lower performance overhead. To validate our algorithm, we fine-tune DeBERTa-v2-XXLarge and Flan-T5 using LoRA adapters on a range of common NLP tasks. We find that P³EFT is competitive with existing privacy-preserving methods in multi-party and two-party setups while having higher accuracy.

## 1   INTRODUCTION

One of the main reasons behind deep learning success is its ability to transfer knowledge between tasks (Tan et al., 2018). When training a model for any particular problem, it is common to reuse previously trained models from other, related problems. In the past, this was typically done by downloading pre-trained model weights from public hubs, then fine-tuning the said models on the downstream task. However, as models grow larger and more compute-intensive, fine-tuning them locally becomes an increasingly difficult task. Furthermore, many recent models are not released, but instead made available as proprietary services.

When a model cannot be fine-tuned locally, many practitioners opt instead for the so-called fine-tuning APIs. These APIs are web services backed by remote servers that host one or several pre-trained models and allow clients to perform limited fine-tuning. More specifically, APIs usually allow their clients to run parameter-efficient fine-tuning (PEFT), such as LoRA (Hu et al., 2022) or Prefix-tuning (Li & Liang, 2021). This is particularly necessary for large language models and image generative models, both of which are notoriously expensive to train.

Most fine-tuning APIs have a single endpoint backed by a pool of servers of a particular organization, such as OpenAI API (OpenAI, 2023) or Hugging Face AutoTrain (Hugging Face, 2023) for fine-tuning language models and Dreambooth API (2023) or OctoAI API (OctoAI, 2023) for fine-tuning diffusion models. Recently, there have also appeared several decentralized fine-tuning systems, such as Petals (Borzunov et al., 2022).

Although the fine-tuning APIs can be convenient, they also introduce new challenges and risks that were absent in local fine-tuning. If a client uses such API to fine-tune the model on sensitive data, they need to ensure that their data will stay private. This is particularly important when dealing with patient's medical records, personal user data or trade secrets. The two main threats to data privacy are that the API provider obtains the private data and that a third party intercepts data in transit. Therefore, data privacy is not guaranteed even if the API provider is trusted. This forces many privacy-sensitive parties to avoid fine-tuning APIs and train their models locally, which is often less efficient and prevents them from using the state-of-the-art models.

In this work, we seek to alleviate this problem by designing a two-party fine-tuning protocol that performs standard parameter-efficient fine-tuning with privacy guarantees. We formulate our protocol as a special case of split learning (or vertical federated learning), where one side (server) holds the pre-trained model and the other (client) has private training data. More specifically, we focus on **the privacy of client's training labels**. While input privacy is also important, we found that inputs can often be anonymized or obfuscated by other means (see Section 2.1).

Instead of developing a specific privacy-preserving architecture or training objective, we seek algorithms that can work with popular existing models and PEFT algorithms. Furthermore, our approach relies on some of the properties of parameter-efficient fine-tuning. Notably, since the adapters are compact, both parties can maintain multiple sets of adapters and swap between them with relative ease. This allows us to design a PEFT-specific algorithm that can solve its task more effectively than general split learning strategies.

We summarize the main contributions of our work as follows:

- We analyze common parameter-efficient fine-tuning algorithms from the perspective of label privacy. We observe that, despite fine-tuning less than $0.1\%$ of model parameters, modern PEFT algorithms leak client's training labels against simple attacks that work for modern pretrained transformers.

- Based on our analysis, we formulate a framework for privacy-preserving parameter-efficient fine-tuning ($P^3EFT$). This framework leverages the properties of PEFT to provably obfuscate the gradients communicated during fine-tuning with no impact on the fine-tuned model quality.

- To verify the practical viability of $P^3EFT$, we conduct experiments on popular real-world PEFT workloads[1]. Notably, we fine-tune DeBERTa-v2-XXL (He et al., 2021) and Flan-T5 (Chung et al., 2022) on a set of standard language understanding problems. We find that, compared to prior split learning algorithms, $P^3EFT$ can maintain label privacy throughout training with significantly smaller accuracy drop.

## 2 BACKGROUND

### 2.1 FEDERATED LEARNING AND SPLIT LEARNING

Privacy preservation in machine learning has been a subject of active study within several frameworks. An important branch of privacy-preserving learning methods is federated learning, or FL (McMahan et al., 2017), which can be broadly described as an approach allowing several parties to train a model jointly without sharing their private data. In particular, vertical federated learning (Hardy et al., 2017; Yang et al., 2019) targets the scenario where different features (including the label) of each training instance are kept by different parties.

One of the most popular approaches to vertical FL for neural networks is split learning (Gupta & Raskar, 2018; Vepakomma et al., 2018), where each party stores its part of the overall model. To train the model in such an approach, it is only necessary to transfer the intermediate activations and the gradients between layers, while the data itself is stored at the premises of the participant hosting each layer. In this work, we focus on the two-party formulation of split learning, where one side stores the features for each example and another one stores the labels.

Recent works have investigated the setting of two-party split learning from the label leakage perspective (Vepakomma et al., 2019; Pasquini et al., 2021): because the label party needs to pass the gradients of the loss function to the non-label party, it is possible for the latter party to deduce the labels by inspecting the gradients or activations or by hijacking the training proecdure. Li et al. (2022) provide a set of attack methods that allow recovering private labels and propose a defense mechanism that injects noise into the gradients; however, they test the approach on pretraining smaller models, and we study finetuning large models on private downstream data.

---

[1]The code is available at `github.com/iclr2023-anonymous/P3EFT`

## 2.2 PARAMETER-EFFICIENT FINETUNING

The majority of large neural networks today are not trained with a specific task in mind: instead, they are pretrained on a general objective and then adapted for the downstream problem. Importantly, the growth in the size of foundation models has led to the increased popularity of parameter-efficient finetuning (PEFT) methods that adapt the model to a given task by training a small number of task-specific parameters. There are several prominent approaches to parameter-efficient finetuning, ranging from trainable prompts (Li & Liang, 2021; Hambardzumyan et al., 2021), to residual adapters (Houlsby et al., 2019; Pfeiffer et al., 2021). We focus on Low-Rank Adaptation (or LoRA, Hu et al., 2022), one of the most popular PEFT methods that adds extra parameters to each weight matrix in the form of a low-rank factorization (see Appendix B for a more detailed description). Such formulation allows LoRA adapters to be merged into the original weights after finetuning; this ability, combined with the simplicity of the method, has made LoRA a broadly popular approach in multiple domains. Still, the approach we propose can be applied to any PEFT method.

Importantly, the connections between data-private learning and parameter-efficient finetuning have been explored in several past works. One of the earlier works at the intersection of these areas is Yu et al. (2022); however, its primary focus is differential privacy, i.e., hiding the identity of each training example rather than hiding the training task itself. As also argued by Li et al. (2022), in the setting of split learning, the non-label party knows the participation of each example in the training procedure; therefore, differential privacy is not applicable in the conditions we study. Zhao et al. (2023) explore the viability of prompt tuning for federated learning and Zhang et al. (2023) study four PEFT algorithms in the setting of *horizontal* federated learning, comparing their task performance, communication costs, and privacy preservation capabilities. The primary distinction between our work and these studies is that we investigate parameter-efficient adaptation in the setting of split learning: instead of training over data split across workers, we aim to finetune a model without disclosing the labels of examples to the model provider.

## 3 PRIVACY-PRESERVING PARAMETER-EFFICIENT FINE-TUNING

In this section, we analyze the privacy of parameter-efficient fine-tuning and propose a protocol for two-party parameter-efficient fine-tuning with the desired privacy guarantees. We begin by analyzing the privacy of API fine-tuning with popular PEFT algorithms in Section 3.1. Then, in Section 3.2, we formulate a protocol for privately computing gradients over fine-tuning APIs. Finally, we formulate the full P³EFT protocol in Section 3.3.

### 3.1 TWO-PARTY SPLIT FINE-TUNING

To analyze the privacy of API fine-tuning, we first need to formulate a common framework for this type of APIs and develop private learning protocols. This step is important, because existing fine-tuning APIs greatly vary in what they offer to the client.

Notably, as of writing of this paper, most API providers ask users to submit their training data, perform fine-tuning with some undisclosed parameters, and returns a handle that can later be used to query the model. This approach offers no avenue for ensuring that client's data is private *from* the provider. Furthermore, this type of API offers clients no flexibility in how they want to perform their fine-tuning.

Another, more flexible type of fine-tuning API allows clients to run individual forward and backward passes over a remote model (Borzunov et al., 2022; Rao et al., 2021; Li et al., 2023). A client can use these APIs to obtain the training gradients for their PEFT adapters, then update adapters with any optimization method. In our work, we adopt this archetype of fine-tuning API as it offers sufficient flexibility to develop privacy-preserving algorithms.

We formulate fine-tuning over an API for two or more parties: a client, and one or several servers. The client owns a training dataset with inputs $X$ and labels $Y$. In turn, each server has the same pre-trained model $h(x_i, \theta) \in \mathcal{R}^d$. Note that the parameters $\theta$ denote not the pre-trained model weights, but the trainable adapter weights for a certain PEFT algorithm. A model can encode an input $x_i \in X$ and produce a $d$-dimensional vector of hidden activations (learned input representations) that depend on the learned adapter weights $\theta$.

To allow fine-tuning, each server offers two API methods: forward$(x, \theta)$ that returns $h(x, \theta)$, and backprop$(x, \theta, g_h) = g_\theta$ that receives gradients $g_h = \frac{\partial L(h(x,\theta))}{\partial h(x,\theta)}$ of an arbitrary loss function w.r.t. model activations and returns the gradients of the same loss function with respect to the specified PEFT parameters, $g_\theta = \frac{\partial L(h(x,\theta))}{\partial \theta}$.

We further assume that both forward$(\cdot)$ and backward$(\cdot)$ APIs are stateless and deterministic, i.e. calling the same API method multiple times (or on multiple servers) with the same inputs produces identical results. Thus, if the model uses dropout or any other form of non-determinism, we assume that clients provide the random seed as a part of $x$.

Real-world fine-tuning APIs are not exactly nondeterministic due to hardware and software limitations. In principle, they can be made exactly deterministic at the cost of slower computation. However, this is not necessary, as our work does not rely on strict determinism up to numeric precision. Finally, fine-tuning APIs can provide several models and offer more than one PEFT algorithm, which we leave out of the scope of our analysis.

To fine-tune a model with this API, a client can initialize adapters locally, alongside with a small task-specific "head", then train both adapters and head on training minibatches. For each minibatch $(x, y) \in D$, a client calls forward$(x, \theta)$ to compute feature representations, then predicts with local "head" and computes task-specific loss function $L$. After that, a client performs backward pass: first, it computes gradients w.r.t. local head inputs $g_h = \frac{\partial L}{\partial h}$, then passes those gradients to a remote server via backward$(x, \theta, g_h)$ API call to compute gradients w.r.t. $\frac{\partial L}{\partial \theta}$. Finally, a client updates both $\theta$ and local "head" parameters using the optimizer of choice.

Before building more advanced algorithms, let us analyze the privacy of client's labels under standard fine-tuning. We consider an "honest, but curious" attacker model. This means that the server will faithfully run the forward and backprop computations as requested by the client without changing the results. Furthermore, we assume that servers are independent and do not communicate client's data between each other. However, a server can recover client's labels by performing arbitrary computations on top of any information it receives from the client.

When training in this way, a client does not directly communicate training labels to the server. However, they do communicate inputs, adapter parameters, and gradients. Furthermore, the server communicates input representations that can be intercepted by a third party.

In Figure 1, we train a DeBERTa-v2-XXL model on the SST-2 sentiment classification dataset. The top row depicts the gradients $g_h$ communicated by the client when calling backprop$(\cdot)$ at different training stages. In the bottom row, we similarly track activations $h(x, \theta)$ that server may compute based on the specified $x, \theta$. We defer further additional figures and details to Section 4.1.

As we can see, both gradients and activations are arranged in such a way that simple k-means clustering would reveal which objects have the same label. The training activations (bottom row) do not reveal labels right away (at least not against this attack). However, they gradually "leak" private

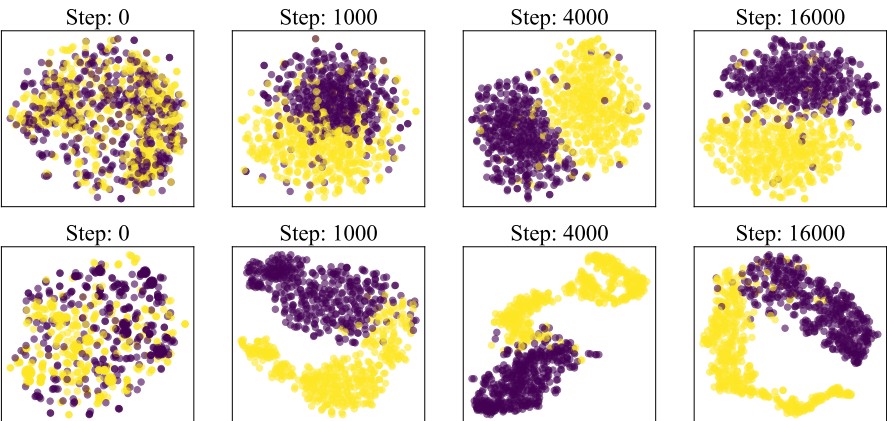

Figure 1: A visualization of top-2 principal components of gradients (top) and activations (bottom) from different fine-tuning steps (left to right). Color indicates the training labels (binary).

label information during training. From an information-theoretic perspective, knowing just one vector of gradients *or* trained activations allows the attacker to learn all but one bit[2] of information about client's private labels.

To summarize, leaving any *one* data source unprotected (gradients, activations or parameters) would already compromise label privacy. However, we found that gradients and activations require different means of protection.

## 3.2 PRIVACY-PRESERVING BACKPROPAGATION

In this section, we formulate an algorithm for "anonymizing" the gradients communicated over a single training step with arbitrary PEFT type. Several prior works approach this by modifying the training objective or model architecture. However, when dealing with a real-world PEFT workload with optimized hyperparameters, changing the model or loss function often results in reduced model accuracy[3]. Thus, we seek an algorithm that preserves both model and training objective.

We design our algorithm based on an observation that **backpropagation is conditionally linear in output gradients**, even when the model itself is nonlinear. Formally, if we take a model $h(\cdot, \cdot)$, a fixed set of trainable parameters $\theta$ and input samples $x$, the backprop "function" computes $\text{backprop}(x, \theta, \frac{\partial L}{\partial h(x,\theta)}) = \frac{\partial L}{\partial \theta}$. For convenience, we shorten it to $\text{backprop}(x, \theta, g_h) = g_\theta$, where $g_h = \frac{\partial L}{\partial h(x,\theta)}$ represents the gradients of some objective function with respect to model activations (outputs), and $g_\theta = \frac{\partial L}{\partial \theta}$ are gradients of the same objective function w.r.t. trainable parameters. In this notation, backprop is linear in terms of $g_h$ for any fixed $x, \theta$.

This becomes self-evident if we view backprop as multiplying $\vec{g}_h$ by the Jacobian of model outputs w.r.t. trainable parameters, $\frac{\partial h(x,\theta)}{\partial \theta}$. If $x, \theta$ are constant, the Jacobian is also constant, and backprop is a linear operator:

$$\text{backprop}(x, \theta, \frac{\partial L}{\partial h(x,\theta)}) = \frac{\partial L}{\partial \theta} = \frac{\partial L}{\partial h(x,\theta)} \times \frac{\partial h(x,\theta)}{\partial \theta} \qquad (1)$$

This observation allows us to design a private backpropagation protocol. To illustrate this protocol, let us first consider a distributed API with two identical independent servers that offer backprop API. Then, for arbitrary vector $\vec{z}$, we can rewrite:

$$\text{backprop}(x, \theta, \vec{g}_h) = \text{backprop}(x, \theta, g_h + \vec{z}) + \text{backprop}(x, \theta, g_h - \vec{z}) \qquad (2)$$

During API fine-tuning, we obtain $\text{backprop}(x, \theta, g_h + \vec{z})$ using an API call to server 1, whereas the second term $\text{backprop}(x, \theta, g_h + \vec{z})$ translates to an API call to server 2. Note that neither of two servers has access to the true gradient $\vec{g}_h$: they only receive the sum $[\vec{z} + g_h]$. If we sample a large noise vector $\vec{z}$ ($\text{Var}(\vec{z}) \gg \|(g_h)\|_2^2$), this sum becomes indistinguishable from noise. However, when both API calls finish, a client can add the result to recover the true $g_\theta = \frac{\partial L}{\partial \theta}$.

If both requests are processed by the same server, it can obviously recover $g_h$ by adding up gradients from both calls, which leads us to the final step. Instead of generating a single noise vector, a client needs to generate (privately) a set of $m > 1$ random vectors $\hat{g}_1, \ldots, \hat{g}_m$ and scalars $\alpha_1, \ldots, \alpha_m$ such that $g_h = \sum_{i=1}^{m} \alpha_i \cdot \hat{g}_i$. Then, for each $\hat{g}_i$, client computes $\text{backprop}(x, \theta, \hat{g}_i)$ as $m$ parallel API calls. Once this is done, client recovers $g_\theta = \sum_{i=1}^{m} \alpha_i \cdot \text{backprop}(x, \theta, \hat{g}_i)$. Note that the client does not reveal scalars $\alpha_1, \ldots, \alpha_m$ to anyone.

This procedure can allow client to safely compute gradients *once*, but, in practice, client usually needs to run many consecutive steps. This creates an additional vector of attack: if the same server receives two sets of parameters $\theta_t, \theta_{t+1}$, they could potentially recover $g_\theta$ by inverting the optimizer.

In the simplest case, if the server somehow knows that the client computes $\theta_{t+1} = \theta_t - \eta \cdot g_\theta$, then they can compute $g_\theta = \frac{\theta_t - \theta_{t+1}}{\eta}$. While $g_\theta$ does not necessarily leak private labels, a server could, in some cases, use $g_\theta$ to recover $g_h$, either fully (e.g. if Jacobian is invertible), or partially.

---

[2]The missing bit corresponds to attacker not knowing which cluster corresponds to label "1".

[3]We validate that experimentally in 4.2

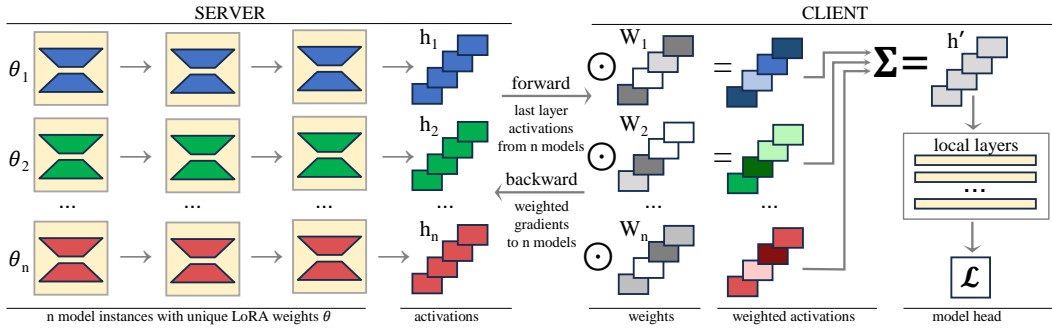

Figure 2: An intuitive illustration of the proposed fine-tuning protocol.

The client has two ways to prevent this attack. The first one is to ensure that no single server runs backprop on two consecutive steps. This is easy to do in decentralized systems where there are many potential servers. However, even when there is a single server, they could be required to set up multiple trusted execution environments (Nvidia, 2023). A more risky alternative is to ensure that the gradients cannot be reversed from consecutive parameters: randomize initial optimizer statistics or add noise to parameters. This solution is easier, but it can adversely affect convergence in some cases. The resulting procedure is formulated in Algorithm 1.

---

**Algorithm 1** private_backprop - Privacy-Preserving backpropagation (from client's perspective)

**Input**: $x$ inputs, $\theta$ adapter weights, $g_h$ gradients w.r.t. activations, $m > 1$ - number of passes
1: $\hat{g}_h^1, \ldots, \hat{g}_h^m, \alpha_1, \ldots, \alpha_m = \text{obfuscate}(g_h, m)$   $\qquad \triangleright$ s.t. $\sum_{j=1}^m \alpha_j \cdot \hat{g}_h^j = g_h$
2: **for** $j = 1, \ldots, m$ **do**
3: $\qquad \hat{g}_\theta^j = \text{backprop}(x, \theta, \hat{g}_h^j)$   $\qquad \triangleright$ server computes $\hat{g}_h^j \times \partial h / \partial \theta$
4: **end for**
5: $g_\theta = \sum_{j=1}^m \alpha_j \cdot \hat{g}_\theta^j$
**Return**: $g_\theta$

---

To summarize, we formulated a procedure that allows a client to compute gradients privately for any given model and PEFT type. Furthermore, since eq. 2 recovers true gradients, this obfuscation method does not affect the training dynamics. However, as we have shown in Section 3.1, gradients are not the only source of privacy leakage.

## 3.3 Full fine-tuning

The other major attack vector are training activations. As the model fits to training data, it's intermediate activations $h(x, \theta)$ allow attackers to recover labels. To combat this issue, we take advantage of the fact that PEFT has few trainable parameters. Instead of learning just one set of trainable parameters, a client creates $n$ independent adapter sets $\theta_1, ..., \theta_n$. Note that this does not require $n$ unique servers: a single server can run multiple sets of adapters. Furthermore, a client can alternate between using different servers for the same adapters. During forward pass, the outputs of different adapters are mixed together using randomized mixing weights $W \in \mathcal{R}^{n,d}$:

$$h'(x, \theta_1, \ldots, \theta_n) = \sum_{i=1}^n W_i \odot h(x, \theta_i) \qquad (3)$$

Overall, we design this model in such a way the combined model $h'$ can predict the labels, but the adapters $h(x, \theta_i)$ do not allow predicting these labels without knowing the mixing weights W. The mixing weights are generated such that initial activations $h'(x, \dots)$ are equal to mean $h(x, \cdot)$ for all $x$. To achieve this, we generate W as follows: first, we generate $n \cdot (n-1)/2$ d-dimensional random vectors $\vec{\xi}_{i,j} \in \mathcal{R}^d \forall i \in [1, n], j \in [i+1, n]$. Then, we add them up in the following way:

$$W = \begin{pmatrix} \frac{1}{n}\vec{e} + \vec{\xi}_{1,2} + \vec{\xi}_{1,3} + \cdots + \vec{\xi}_{1,n} \\ -\vec{\xi}_{1,2} + \frac{1}{n}\vec{e} + \vec{\xi}_{2,3} + \cdots + \vec{\xi}_{2,n} \\ \cdots \\ -\vec{\xi}_{1,n} - \vec{\xi}_{2,n} - \vec{\xi}_{3,n} - \cdots + \frac{1}{n}\vec{e} \end{pmatrix} \qquad (4)$$

Here, $\vec{e}$ stands for a vector of all ones. The purpose of these mixing weights is to ensure that the gradients w.r.t. individual $h(x, \theta_i)$ are obfuscated, but the averaged model behaves the same as regular PEFT adapter. To illustrate this, consider $n=2$ identical LoRA adapters $\theta_1, \theta_2$. During the first training step $h(x, \theta_1) = h(x, \theta_2)$. Therefore,

$$h'(x, \theta_1, \ldots, \theta_n) = (1/2\vec{e} + \vec{\xi}_{1,2}) \odot h(x, \theta_1) + (1/2\vec{e} - \vec{\xi}_{1,2}) \odot h(x, \theta_2) = h(x, \theta_1) \quad (5)$$

However, the two adapters will learn different functions as they receive different gradients. From the first update on, $h'$ will be equal to an average of adapter predictions.

Finally, to ensure that individual adapters $h(x, \theta)$ do not accidentally "learn to leak" labels, we maintain this over the course of training with a privacy regularizer inspired by Ganin & Lempitsky (2015). This ensures that it is impossible to predict labels from individual adapters $h(x, \theta_i)$.

Intuitively, on each training step, client fits $n$ linear "heads" that learn to predict labels $y$ from $h(x, \theta_i)$, then performs an adversarial update of $\theta_i$ to prevent the "head" from predicting $y$.

Formally, each of $n$ "heads" minimize the same objective function as the full model. For instance, if the full model solves multi-class classification, each head is trained to minimize cross-entropy: $\eta_i^* = \arg\min_{\eta_i} \sum_{x,y \in D} -y \cdot \log \frac{e^{\langle \eta_{ij}, h(x, \theta_i) \rangle}}{\sum_k e^{\langle \eta_{ik}, h(x, \theta_i) \rangle}}$, where y is one-hot encoding of the correct class.

The whole adversarial update takes place locally on client's side, using the same $h(x, \theta)$ it uses for the main training objective. The resulting procedure appears complicated but it typically takes negligible time compared to running the large pre-trainied model $h(x, \theta)$. Furthermore, since adversarial "heads" are linear, minimizing the objective above is done with standard logistic regression solver.

To summarize, our approach combines the two proposed ideas: we use the private backpropagation algorithm from Section 3.2 to protect the gradients, then trains a mixture of adapters in such a way that obfuscates learned activatons leaking labels. The resulting procedure is described in Algorithm 2. In the next section, we will evaluate the efficacy of P³EFT on popular NLP benchmarks.

## 4 EXPERIMENTS

The main goal of this study is to find a practical method of private fine-tuning that would scale to modern pre-trained transformers. To verify if P³EFT meets these criteria, we chose to evaluate it not on typical datasets used in split-learning (e.g. CIFAR10, Krizhevsky (2009)), but on fine-tuning recent pre-trained transformers on NLP bechmarks representative of real-world tasks.

To that end, we chose two pre-trained models: DeBERTa-XXLarge (He et al., 2021) and Flan-T5-Large (Chung et al., 2022). We train these models to perform sentiment classification on SST-2 (Socher et al., 2013) and paraphrase identification on MRPC (Dolan & Brockett, 2005), both of which are parts of the GLUE benchmark (Wang et al., 2018). For each model, we train LoRA adapters with rank 8. To improve reproducibility, we reuse the recommended hyperparameters from Hu et al. (2022) for the two corresponding tasks.

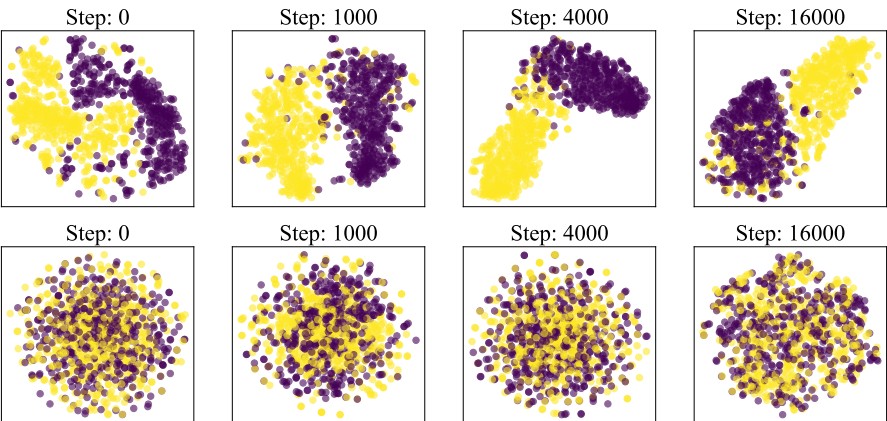

Figure 3: Gradients of cross-entropy w.r.t. LoRA parameters for DeBERTa-v2-XXLarge. The top row corresponds to normal backpropagation and the bottom row uses privacy-preserving backprop.

### 4.1 PRIVACY OF GRADIENTS AND ACTIVATIONS

For this experiment, we train DeBERTa-XXLarge on SST-2 dataset using a regular LoRA adapters. First, we train the model locally and track model activations $h$ and gradients w.r.t. those activations. We apply principal component analysis to them into 2-dimensions and visualize them in Figure 1. Similarly, we visualize gradients of individual per-sample loss functions w.r.t. LoRA parameters $\theta$ in Figure 3 (top row). As we mention earlier, a hypothetical attacker could easily recover private labels by performing K-Means clustering over any data source: activations, gradients w.r.t. activations, and as well as individual gradients w.r.t. parameters.

Next, we run the same experiment using privacy-preserving backpropagation as defined in Section 3.2. We use $n = 2$ with noise variance set to 1000. As expected, we observed the same learning curve as with normal training. However, instead of sending gradients w.r.t. activations to the server, client uses a specially crafted random noise vectors that are not informative. In Figure 3(bottom) we plot the same kind individual gradients as in the top row, except that we visualize the gradients computed by the first of the two servers. Finally, we train XGBoost (Chen & Guestrin, 2016) with default hyperparameters to predict labels given the noisy gradients (pre-PCA): the resulting classifier is able to fit the training data perfectly, but has at most $50.4\%$ accuracy on a balanced test set.

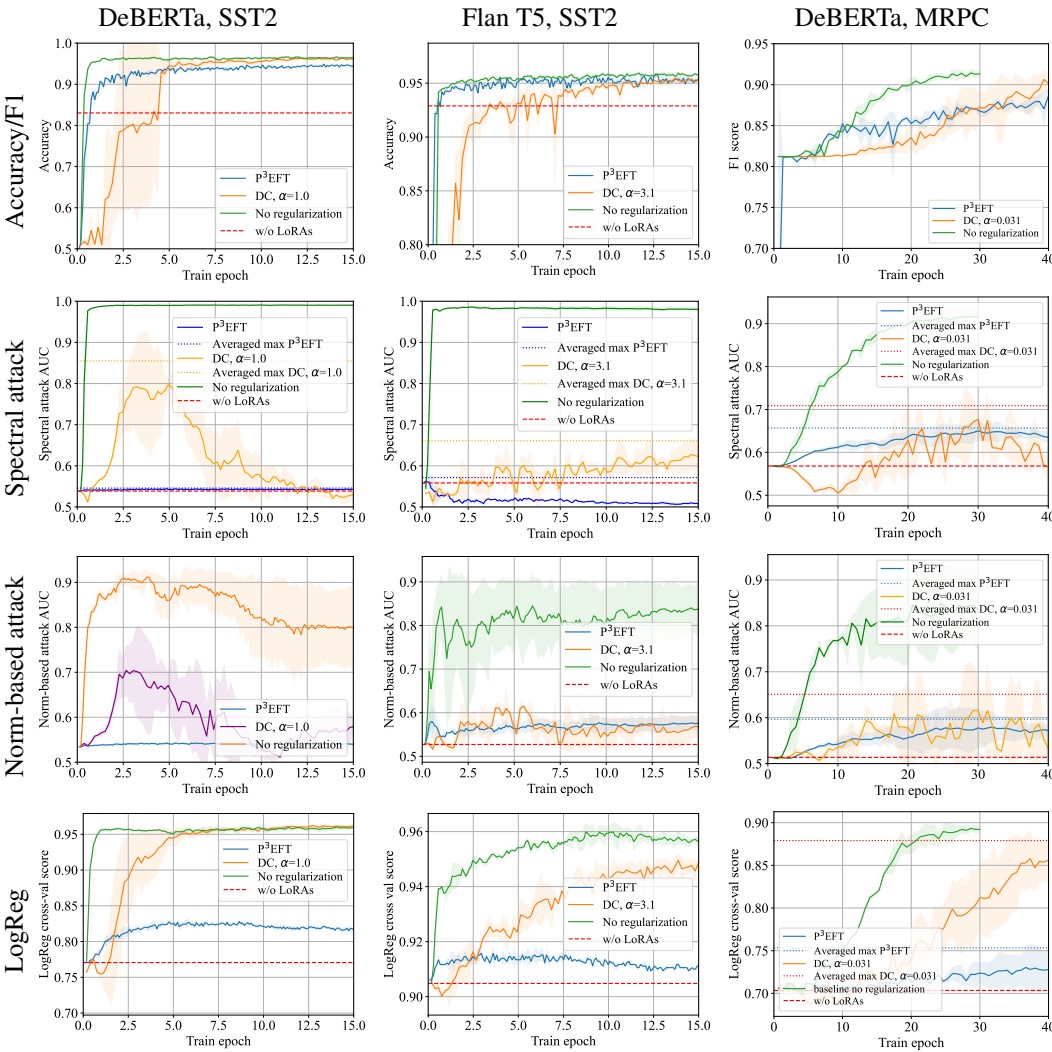

Figure 4: Combined PEFT accuracy and privacy evaluations. See detailed description in Section 4.2.

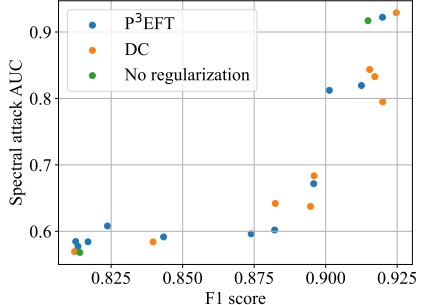 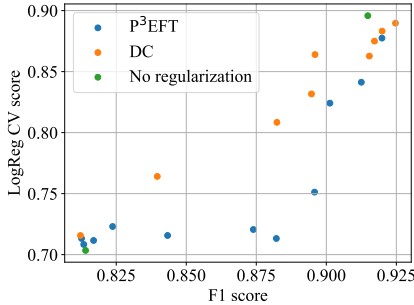

Figure 5: Combined sensitivity charts for DeBERTa-xxlarge with MRPC.

## 4.2 MAIN FINE-TUNING EXPERIMENTS

Next, we evaluate the full P³EFT algorithm in the same setting. To control for task and model type, we consider three fine-tuning setups: DeBERTa-v2-XXLarge on SST-2, DeBERTa-v2-XXLarge on MRPC, and Flat-T5-Large on SST2. For each setup, we compare against three baselines:

- **Distance Correlation (DC).** Our re-implementation of the distance correlation defense formulated in (Sun et al., 2022). For this baseline, we tune $\alpha$ separately for each task. We tune $\alpha$ to maximize accuracy with a constraint that DC has same or comparable privacy as our algorithm.

- **Training w/o LoRA adapters.** In this baseline, the client gathers $h$ activations once at the beginning, with no adapters, then proceeds to train local "head" layers on top of said activations. As a result, the algorithm cannot leak information about training labels except for what is stored in X.

- **Training LoRA with no regularization** refers to training a single LoRA adapter normally. This baseline represents an upper bound on model accuracy, but lacks privacy.

For each algorithm, we report task-specific metric (Accuracy or F1) as well as 3 privacy measures:

- **Spectral attack** - vulnerability to attack proposed in Sun et al. (2022), measured as classifier ROC AUC, lower is better privacy.

- **Norm attack** - vulnerability to a variant of attack proposed in Li et al. (2022), measured as classifier ROC AUC, lower is better.

- **LogReg** - the cross-validation accuracy of logistic regression that was trained to predict class labels. Pessimistic estimate of privacy. Lower is better privacy.

We report main fine-tuning results in Figure 4. Overall, P³FT algorithm achieves nearly the same accuracy and outperforms Distance Correlation-based algorithm in terms of accuracy given the same privacy level. However, both P³FT and DC can achieve different accuracy-to-privacy trade-offs depending on the value of the regularizer coefficient. To explore this, we also conduct sensitivity experiments where we vary the regularizer coefficients of both algorithms and report our findings in Figure 5. While both algorithms offer a wide range of configurations, P³EFT offers slightly better trade-offs. We evaluate additional hyperparameter configurations in Appendix D

## 5 CONCLUSION

In this work, we analyze privacy-preserving fine-tuning of large neural networks in the context of parameter-efficient fine-tuning and the two-party split learning setting. We show that while standard fine-tuning suffers from label leakage even in the parameter-efficient case, it is possible to leverage the efficiency of PEFT to alter the procedure without any significant performance drawbacks. We test the resulting method, named P³EFT, on a range of pretrained language models and multiple datasets, showing that it is competitive with a strong baseline in terms of label privacy while having higher task performance. In future work, it might be possible to explore alternative ways of using parameter-efficient fine-tuning to preserve privacy.

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

## A  FORMAL ALGORITHM DEFINITION

Below, we define the full P³EFT algorithm. In Algorithm 2, main_loss is the task-specific objective e.g. cross-entropy; reg_loss is the adversarial regularizer described in Section 3.3. We denote client-side model "head" as $f(h, \psi^t)$, where $\psi$ represent trainable head parameters. Finally, opt_step function performs a single gradient descent step with a task-specific optimizer, typically Adam (Kingma & Ba, 2014).

## B  INFORMAL DESCRIPTION OF LORA FINE-TUNING

For convenience, we provide a brief summary of fine-tuning with LoRA (Hu et al., 2022). This PEFT method was originally designed for fine-tuning large pre-trained language models on downstream NLP tasks. These language models are typically based on the Transformer architecture (Vaswani et al., 2017), where most trainable parameters are allocated to linear layers in multi-head self-attention and feedforward blocks.

---

**Algorithm 2** P$^3$EFT - full training algorithm

---

**Input**: dataset $D = \{X, Y\}$, $n > 1$ number of adapters, $\alpha \geq 0$ - regularizing weight, $m > 1$ number of obfuscated gradients

1: Initialize head $\psi^0$, mixing weights $W_i$ and adapters $\theta_i^0, i = \overline{1, n}$
2: **for** $t = 0, 1, \ldots, T - 1$ **do**
3:      Sample batch $\{x^t, y^t\}$
4:      **for** $i = 1, \ldots, n$ **do**
5:          $h_i^t = h(x^t, \theta_i^t)$                                    ▷ by server
6:          $l_i = \text{reg\_loss}(h_i^t, y^t)$                             ▷ by client
7:      **end for**
8:      $h' = \sum_{i=1}^{n} W_i \odot h_i^t$
9:      $l = \text{main\_loss}(f(h', \psi^t), y^t)$
10:     $L = l + \alpha \cdot \sum_{i=1}^{n} l_i$
11:     **for** $i = 1, \ldots, n$ **do**
12:         $g_h = \partial L / \partial h_i^t$                  ▷ Client performs partial backprop locally
13:         $g_i^t = \text{private\_backprop}(x, \theta_i^t, g_h, m)$
14:         $\theta_i^{t+1} = \text{opt\_step}(\theta_i^t, g_i^t, t)$
15:     **end for**
16:     $\psi^{t+1} = \text{opt\_step}(\psi^t, \partial l / \partial \psi^t, t)$
17: **end for**
     **Return**: $\psi^T, \theta_1^T, \ldots, \theta_M^T$

---

In the first stage of LoRA fine-tuning, user augments the model with adapters. To do so, a user goes over linear layers in transformer blocks and adds two trainable matrices, $A$ and $B$ that affect this layer's forward pass. Let $W_i \times x + b_i$ be the original layer with $n$ inputs and $m$ hidden units. Here, $W_i \in \mathcal{R}^{m \times n}$ is a pre-trained weight matrix, $b_i \in \mathcal{R}^m$ is a pre-trained intercept vector and $x \in \mathcal{R}^n$ represents a vector of inputs to this particular layer. During the forward pass, a layer with LoRA adapter computes $W_i \times x + b_i + B_i \times A_i \times x$, or equivalently, $(W_i + B \times A) \times x + b_i$. Here, $A_i$ and $B_i$ are two newly added matrices that constitute a LoRA adapter.

The adapter matrices $A \in \mathcal{R}^{r \times n}$ and $B \in \mathcal{R}^{m \times r}$ have a very small intermediate dimension $r$. For instance, when training GPT-3 with LoRA adapters, authors use $1 \leq r \leq 64$, whereas the main weight dimensions are $m = n = 12288$. The first matrix $A$ is initialized with small random normal values, and the second matrix $B$ is initialized at zeros. That way, initial $A$ and $B$ do not affect the model predictions.

Once all adapters are initilized, the user trains all $A_i$ and $B_i$ matrices of the model, while keeping the rest of the weights frozen. This way, only a small faction (less than 1%) of model weights are updated. Once the training is over, the learned adapters $A_i$ and $B_i$ can be merged into the main weights ($W_i := W_i + A_i \times B_i$) or used separately.

LoRA adapters are designed with two objectives in mind: i) to allow fine-tuning models in limited GPU memory and ii) to allow inferencing many fine-tuned models using one inference server. When fine-tuning, LoRA achieves small memory footprint due to the fact that user does not need to compute gradients (or optimizer statistics) for billions of main model parameters. During inference, a server can keep a library of several adapters for different tasks and swap between them on demand.

## C   OFFSITE-TUNING

One alternative to private LLM fine-tuning is Offsite-Tuning, described in Xiao et al. (2023). This approach involves conducting all computations on the client side, making it much less vulnerable to potential attacks. To achieve this, the server constructs a smaller version of the main model and transfers it to client.

This smaller model typically contains a subset of original model layers (e.g. see Uniform Layer-Drop algorithm in Xiao et al. (2023)) and is fine-tuned using knowledge distillation. The resulting small model consists of three parts: i) several first layers of the main model ii) several last layers and iii) a distilled emulator all remaining layers.

Once the client receives the resulting model, it runs local fine-tuning on the private dataset. During this fine-tuning stage, a client only updates the first and last layers of the model, keeping the emulator frozen. Once the training is done, the client transfers the updated layers back to the server, where they can be inferenced with the rest of the main model.

The main limitation of this approach is that the distilled model still needs to be fairly large. Xiao et al. (2023) needs the smaller model to be at least one-third of the main model size to achieve competitive fine-tuning accuracy. As a result, the client needs to expend compute on the same order of magnitude as when fine-tuning the original model. In contrast, API fine-tuning only requires the client to perform forward and backward passes through the model "head", which typically is typically a small MLP. Therefore, a client will be able to perform API fine-tuning using cheap general purpose hardware, e.g. a laptop.

## D  ADDITIONAL EVALUATIONS

In this section, we report additional fine-tuning results similar to Figure 4, but with more hyperparameter configurations for some baselines. The results are presented in Figure 6.

## E  REPRODUCIBILITY STATEMENT

### E.1  OBJECTIVE AND SCOPE

This paper introduces P$^3$EFT as a way to fine-tune models over an API while keeping the labels private. We believe that our results are reproducible by other researchers and practitioners to validate them and use in subsequent research and applications.

### E.2  METHODOLOGY AND ALGORITHM

We have described the P$^3$EFT method in Section 3.3. This should help anyone interested in understanding and trying out the proposed technique.

### E.3  EXPERIMENTAL SETUP

Model Architectures: We used DeBERTa and FlanT5 models, which anyone can access for research.

Datasets for Calibration: We used the commonly available for research and testing datasets from GLUE Wang et al. (2018): SST-2 and MRPC.

Evaluation Metrics: We measured training quality using common for these tasks metrics and measured privacy preservation using leak AUC approach established in this research area as presented in Li et al. (2022). We calculated the value of this metric for 2 types of attacks: spectral attack Sun et al. (2022) and norm based attack Li et al. (2022). In addition, we took the accuracy value of attacks via KMeans as the baseline.

Comparison Baselines: We compared our method to method based on distance correlation from Sun et al. (2022). Software: We used popular open-source packages like PyTorch and Transformers (see the full list in the repository requirements.txt file).

Hardware: We used Nvidia GPUs including A100.

### E.4  SENSITIVITY AND ABLATION STUDIES

We have listed all the settings we used for P$^3$EFT experiments and conducted thorough studies to check how different parts of our method affect the results. This should help in understanding and verifying how our method works. We tried different random seeds to make sure our method is robust, giving more confidence that you'll get similar results when trying it out.

### E.5 CODE AND IMPLEMENTATION

You can find all our code for P³EFT at `github.com/iclr2023-anonymous/P3EFT`, open for anyone to use. We have included instructions on how to run experiments and checked the results, making it easier for anyone interested to follow along.

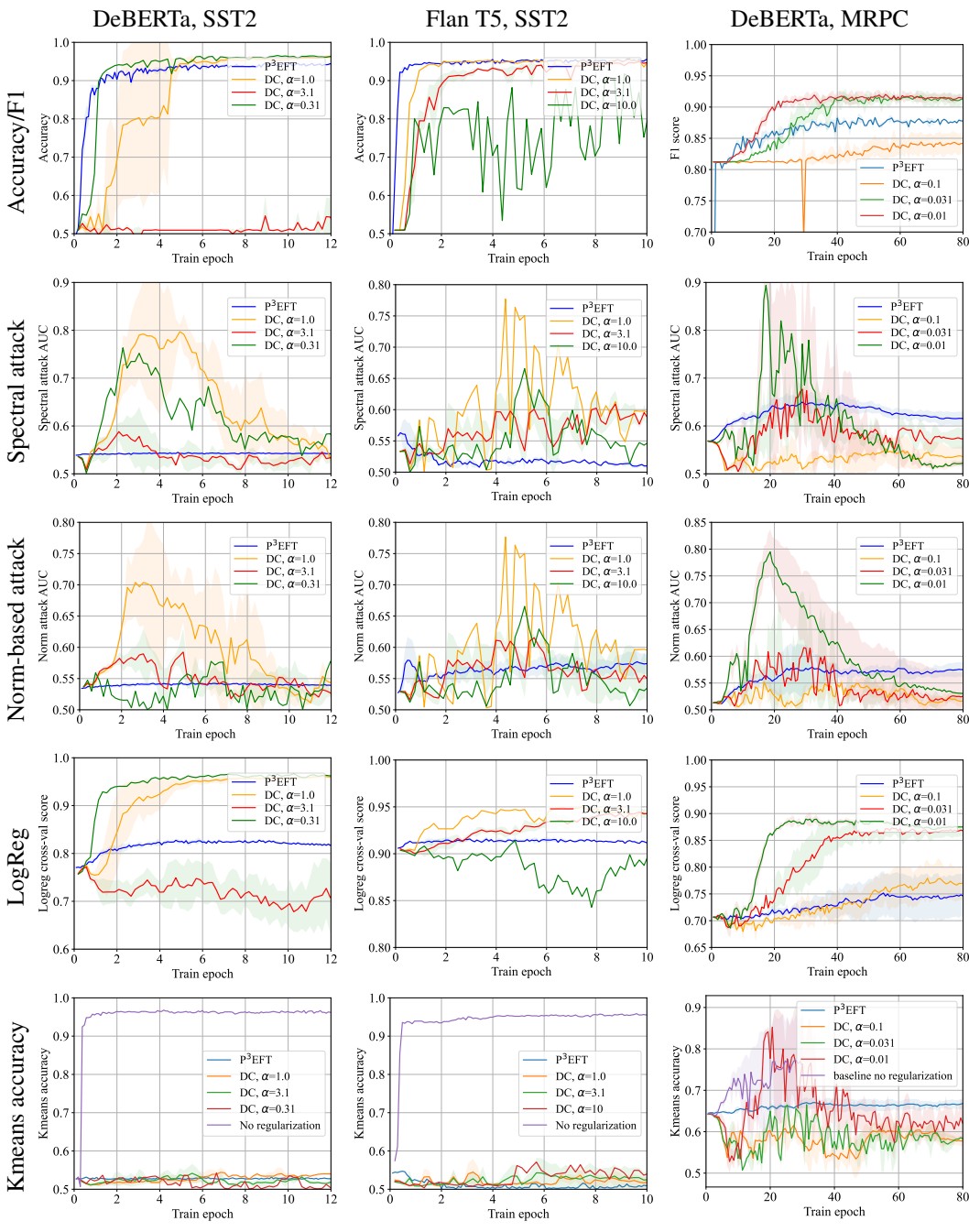

Figure 6: Combined PEFT accuracy and privacy evaluations with additional hyperparameter configuration. See detailed description in Section 4.2 for metric descriptions.

### E.6 ACCESSIBILITY AND LIMITATIONS

The public availability of training code, configurations and datasets should help in verifying our main findings about P$^3$EFT label privacy preservation performance. However, there might be minor inconsistencies due to parallel execution in CUDA, but we believe the details provided are clear enough to reproduce the main findings of our paper.

### E.7 CONCLUSION

The detailed explanation and resources provided in this paper should make it easy for anyone to verify and build upon our work. We hope this contributes to the ongoing research in privacy preservation in large language models.

