# OpenReview forum: "Privacy Preserving API Fine-tuning for LLMs"
_ICLR.cc/2024/Conference — Submitted to ICLR 2024_

### Official Review · Reviewer_awQz · 2023-10-22

**Soundness:** 1 poor
**Presentation:** 3 good
**Contribution:** 2 fair
**Rating:** 3
**Confidence:** 4

**Summary:**

Fine-tuning LLMs via API is a new trend in which users send their data to a server and let the server do the fine-tuning. This paper assumes the samples are pairs of features and labels \{(x, y)\}, and studies how to protect the privacy of the labels. The server is assumed to provide two API functions. The first is a forward function that returns activations ($h$). After receiving the activations, users compute the loss ($l$) by themselves and send $\partial l/\partial h$ to the server. The server then use the second backward API function that uses $\partial l/\partial h$ to do backpropagation to compute the gradients. The authors propose two empirical ways to prevent the server from inferring the labels from activations and gradients. Unfortunately, I have several concerns regarding the protection effectiveness.

**Strengths:**

1.Privacy-preserving API fine-tuning is an important problem and is very challenging due to the two-party learning nature. This paper provides some preliminary exploration towards solving this problem.

2.The ideas borrow some insights from the secure multi-party aggregation literature and are intriguing.

**Weaknesses:**

1.Regarding the privacy-preserving backpropagation. Although $\partial l/\partial h$ is protected, the server still has clean $\partial h/\partial \theta$. This still leaks information about the label. The 2-dimensional TSNE of $\partial h/\partial \theta$ should be similar to $\partial l/\partial \theta$, because multiplying $\partial l/\partial h$ is only a linear operator.

2.Regarding the privacy-preserving forward. If the label can be predicted via only linear transformations of $h_i$, then it means the unaggregated $\{h_i\}$ leaks a lot of information about the label. E.g., the server can simply run some clustering algorithms, which will achieve good performance because $\{h_i\}$ are linearly separable w.r.t. the labels.

**Questions:**

See above.

---

> ### Author Response · Authors · 2023-11-21
>
> Thank you for the review! Please find our response to your comments below:
>
> > Regarding the privacy-preserving backpropagation. Although $\partial l / \partial h$ is protected, the server still has clean $\partial h/\partial \theta$. This still leaks information about the label. The 2-dimensional TSNE of $\partial h/\partial \theta$ of should be similar to $\partial l/\partial\theta$, because multiplying is only a linear operator.
>
> For this concern, we agree that $\partial h/\partial \theta$ is similar to $\partial l/\partial\theta$. For this reason, we report $\partial l/\partial\theta$ in Figure 3 ( in addition to $\partial l / \partial h$ before that). There, we observe that $\partial l/\partial\theta$ is also obfuscated through private backpropagation. Please also note that while $\partial l/\partial\theta$ and $\partial h/\partial\theta$ are indeed linearly dependent, there is a  distinct linear dependency **for each sample and at each training step**: as a result, simply trying to recover them with TSNE is unlikely to be fruitful.
>
> If there is any specific experiment we can run to check your hypothesis, we are happy to discuss that.
>
>
> > Regarding the privacy-preserving forward. If the label can be predicted via only linear transformations of $h_i$, then it means the unaggregated $h_i$ leaks a lot of information about the label. E.g., the server can simply run some clustering algorithms, which will achieve good performance because $h_i$ are linearly separable w.r.t. the labels.
>
> We split this concern into two statements:
>
> 1. If the label can be predicted via only linear transformation, then $h_i$ leaks information about the label
>
> 2. If 1. is true, the server can run clustering that will achieve good performance for separating labels
>
> The first statement is true: **if** the label can be predicted via linear transformation, then $h_i$ leaks labels **in the information-theoretic sense**. However, in our case **the label cannot be predicted with a linear transformation**. The algorithm described in Section 3.3 explicitly trains $h_i$ so that no fixed linear transformation can recover labels from them. This idea is formulated and discussed in the first half of page 7.
>
> Furthermore, even the label could be predicted linearly, this would not guarantee that a clustering algorithm would be able to recognize it. To illustrate this point, consider a uniform ball of data points with a radius of 1 that contains points with two labels. All points with label 1 are above a certain (e.g., random) separating hyperplane, and all points with label 0 are below that hyperplane. Therefore, the data is linearly separable. However, no clustering algorithm can distinguish between the correct hyperplane and any other hyperplane without knowing labels. Without labels, the data is a uniform ball that can be split equally well in any direction.
>
> In preliminary experiments, we found that 2-cluster k-means has near-random performance for both P$^3$EFT and baseline methods. We will add these results in the revised version of the paper.

---

### Official Review · Reviewer_DV4T · 2023-10-29

**Soundness:** 1 poor
**Presentation:** 2 fair
**Contribution:** 1 poor
**Rating:** 5
**Confidence:** 3

**Summary:**

The authors search for a way to fine-tune models over an API while keeping the labels private. The authors analyze the privacy of popular algorithms for parameter-efficient fine-tuning when training over an API.

**Strengths:**

- The topic of privacy-preserving LLM is timing and essential.

**Weaknesses:**

- No clear security model.
- The idea seems to be wrong.

**Questions:**

The reviewer has major concerns about the correctness of the idea.

- In Section 3, the authors claimed that "formulate a protocol for two-party" in the 1st sentence. In the abstract, "the client ..., and the server ...." A client and a server constitute "two-party" already. However, Equation 2 in Section 3.2 contains "two identical independent servers that offer backprop API." The number of parties is not corresponding.

- As for the formulation, it looks like an application of the $n$-out-of-$n$ secret-sharing scheme. In particular, Equation 2 is essentially similar to Part 2 in [REF1]. Additionally, secret-shared backpropagation has already been solved in the early work [REF2].

[REF1] https://www.cs.columbia.edu/~tal/4261/F19/secretsharingf19.pdf

[REF2] Mohassel, Payman, and Yupeng Zhang. "Secureml: A system for scalable privacy-preserving machine learning." 2017 IEEE symposium on security and privacy (SP). IEEE, 2017.

Could the authors explicitly formulate the security model?
Could the authors explain the difference between the proposed formulation and secret sharing?

**Details Of Ethics Concerns:**

N.A.

---

> ### Author Response · Authors · 2023-11-21
>
> We thank the reviewer for their feedback; we do our best to address their concerns below. We also ask the reviewer for clarifications on several replies.
>
> > No clear security model.
>
> We firmly believe that we have provided a detailed portrayal of the attacker model in section 3.1 (i.e., “honest, but curious” model with details); however, we are open to improving it further. If there are any specific aspects that require clarification, kindly provide further details so that we can address them in the revised version.
>
> > Equation 2 in Section 3.2 contains "two identical independent servers that offer backprop API."
>
> Thank you for pointing out this oversight! We will update the submission to clarify that we consider two setups: one with a single server and the other with multiple servers. We will also specify which setup we are addressing in each specific part of the article.
>
> > In particular, Equation 2 is essentially similar to Part 2 in [REF1].
>
> We thank the reviewer for drawing our attention to the technique of secret sharing, and we agree that it has similarities with Equation 2. However, we believe that there is still a significant difference between these methods. Please see the upcoming General Response for details.
>
> > Additionally, secret-shared backpropagation has already been solved in the early work [REF2].
>
> We agree that the algorithm proposed in [REF2] does address the challenges of private forward and backward passes in the case of two servers. However, in our opinion, their solution may not be suitable for the scenario of fine-tuning models via APIs. This article uses MPC-friendly approximation of softmax, but trains the model from scratch. Nevertheless, the original model was pretrained with original softmax, and in order to fine-tune it using their method, it would be necessary to replace softmax with an MPC-friendly approximation. However, in general, replacing a layer does not guarantee equivalent results, and the authors of the paper did not specifically investigate this particular scenario of fine-tuning.
>
> Furthermore, in their case, the algorithm always requires **two servers**, whereas our method, with slight modifications, operates effectively with **just one server** (second part of section 3.2).

---

### Official Review · Reviewer_7KRM · 2023-10-31

**Soundness:** 3 good
**Presentation:** 3 good
**Contribution:** 2 fair
**Rating:** 5
**Confidence:** 5

**Summary:**

This paper proposed to fine-tune models over an API with privacy requirement on labels. Under a parameter-efficient fine-tuning framework, the paper analysed the possible ways the label information can be leaked, i.e., from gradients or intermediate activations. Experiments justified that the proposed method can defend against recent attack studies.

**Strengths:**

This paper is well presented and the targeted privacy-preserving in tuning/training is an interesting research topic. The authors have reviewed some recent advanced works, especially the ones related LLMs. The methodology is clearly stated, and the experimental results are basically convincing.

**Weaknesses:**

My major concerns are two folds; one is the practical significance of the problem setting and connection to some related topics, and the other concern is novelty of methodology. Please see my detailed comments under Questions.

**Questions:**

1.	From my understanding, the connection of the problem setting with vertical federation learning is contrived in terms of predictive tasks. But I agree that in some scenarios, labels are valuable and privacy preserving might be necessary. In this sense, how about local differential privacy on labels or noisy label learning? Because they are also regarded as solutions to preserving labels. There should be at least some discussion on telling the readers what the advantages of the proposed method are over these existing strategies.
2.	Following 1, with access to the full features of target domain, this work is also related to source-free domain adaptation. I understand the applied loss takes label in this work and thus should be more informative than UDA. It would be better if the necessity of using labels could be clarified.
3.	There is not much referring to the “local layers” in Fig. 1. Are these layers learnable or fixed? Can you explain why it is rational to be learnable/fixed for clients in real scenarios?
4.	When taking about fine-tuning APIs in the paragraph 3-4, I think some recent works are missing, especially from the privacy preserving motivation.

       [1] Earning Extra Performance from Restrictive Feedbacks, 2023
       [2] Offsite-Tuning: Transfer Learning without Full Model, 2023

5.	From my understanding, the technique on gradient privacy preserving is based on zero-order optimization and the random weights for activation is like a code book maintained locally. Can you explain what the differences/novelties are compared to previous work in terms of the two techniques?
6.	If an adversary knows how $ z$ is sampled and $g_h$ could be exposed via sum even the norm of $z$ is large. Noticed n parallel calls has been used as a workaround, it would be better if the cost and benefits trade-off is provided.
7.	Presentation issues. The last paragraph of page 3, $h$ is not well presented. $h’$ is used in Fig. 1 while it is $h^*$ in the main text.

---

> ### Author Response · Authors · 2023-11-21
>
> > In this sense, how about local differential privacy on labels or noisy label learning?
>
>
> Thank you for pointing out this oversight. We fully agree that the paper will benefit from including a discussion and comparison of our method with techniques based on noisy labels and differential privacy.
>
> Previous work has shown that gradients with respect to activations are a very vulnerable aspect of split learning in terms of privacy. Techniques such as noisy labels and differential privacy are used to overcome this very problem [1,2]. However, these techniques rely on using noisy gradients instead of the true gradients, which can lead to a decrease in final quality (see e.g. non-private baselines in [3], [4]).
>
> In contrast to previous work, parameter-efficient fine-tuning does not require computation of gradients with respect to **all parameters**. We only need to update a very small **subset of parameters**, therefore it is needed for the client to know true gradients just for that subset. This allows us to compute true gradients in a completely private manner without decreasing final quality.
>
>
> > Following 1, with access to the full features of target domain, this work is also related to source-free domain adaptation. I understand the applied loss takes label in this work and thus should be more informative than UDA. It would be better if the necessity of using labels could be clarified.
>
>
> The majority of language models have been pretrained either on a masked word prediction or causal language modeling objective. As a result, their pretrained activations are not suitable for classification tasks such as sentiment classification or predicting logical relations between statements. We can provide one of our experiments as evidence for this claim: as a baseline, we trained only the classification head without adapters, and the results were significantly worse (Figure 4). Thus, we believe that unsupervised learning may not be very effective for this task. If you know of any results that show otherwise, we would greatly appreciate it if you could point us to them so that we can compare our method with them.
>
>
>
> > There is not much referring to the “local layers” in Fig. 1. Are these layers learnable or fixed? Can you explain why it is rational to be learnable/fixed for clients in real scenarios?
>
>
> These layers are learnable, they correspond to a standard learnable classification head for BERT fine-tuning. Running those computations takes negligible time compared to LLM and thus can be performed even by general purpose computers without dedicated accelerators (e.g. GPUs). Therefore, we believe that there is no need to freeze these layers as it would reduce the number of trainable layers and thus the overall model capacity. Additionally, to the best of our knowledge, training only the head is a standard solution for fine-tuning BERT for classification tasks.
>
> > When taking about fine-tuning APIs in the paragraph 3-4, I think some recent works are missing, especially from the privacy preserving motivation.
>
>
> We appreciate your bringing these articles to our attention. We acknowledge that both of these methods are potentially much less vulnerable to potential attacks than our method, but they do have some drawbacks which we will discuss below.
>
> From our perspective, the first article presents a promising approach for our problem setting, but there may be issues with its application due to the large number of forward and backward passes required to obtain gradient estimates, which increases compute cost.
>
> The second article also proposes an elegant method that is fully private for the client by design. This method still requires running a distilled model on the client side. However, the authors explicitly state in their limitations that they used a distilled model size that was not smaller than one-third of the size of the original model. This is still incredibly challenging for clients due to the large memory and computational resources required for training. In contrast, our method only requires the client to perform forward and backward passes through the head layers, which poses no problem. Therefore, we believe that our method is potentially more scalable.

---

> > ### Author Response · Authors · 2023-11-21
> >
> > > From my understanding, the technique on gradient privacy preserving is based on zero-order optimization and the random weights for activation is like a code book maintained locally. Can you explain what the differences/novelties are compared to previous work in terms of the two techniques?
> >
> >
> > To clarify, we do not use zero-order optimization methods. In our approach, we use gradient-based methods (specifically, we optimize the weights using Adam). The procedure outlined in Section 3.2 is designed to give the client opportunity to compute true gradients of the loss with respect to the adapter weights in a way that prevents the server from reconstructing these true gradients (server sees only noisy gradients). We apologize if our explanation was unclear and we will strive to improve it in a revised version.
> >
> > Regarding the reference to "the random weights for activation", as we understand it, you are referring to the random weights that are multiplied with the activations. These weights remain frozen throughout the entire procedure. We use these weights to ensure that different activation components from different copies of the model (which only differ in adapters) contribute differently to the overall sum, which is used as the input for the head. For example, the first activation component from the first copy contributes the most to the first component of the sum, the second activation component from the second copy contributes the most to the second component of the sum, and the third component of the sum receives an equal contribution from the third components of both the first and second copies, and so on. Intuitively, this allows the distribution of activations for each individual model copy to be more entangled, making it harder to predict labels based on activations alone. If this idea has been used in prior works, please let us know.
> >
> > > If an adversary knows how $z$ is sampled and $g_h$ could be exposed via sum even the norm of is large. Noticed n parallel calls has been used as a workaround, it would be better if the cost and benefits trade-off is provided.
> >
> >
> > This line of reasoning would work for the simplified two-party equation (2), if we decide to apply that formula to a single server instead. Please note that this formula is meant as a simple demonstration for two independent servers, each of which only knows one noisy gradient. We further extend this formula to multiple servers in the two paragraphs following (2). “N parallel API calls” are also not necessary for our method and were mentioned more for the purpose of demonstrating that we simply need a set of vectors such that the true gradients belong to their linear span. In the future, we will strive to clearly separate the overall narrative and the final result.
> >
> > > The last paragraph of page 3, $h$ is not well presented. $h'$ is used in Fig. 1 while it is $h*$ in the main text.
> >
> >
> > Thank you very much for your feedback, we will definitely address this discrepancy.
> >
> > [1] Label Leakage and Protection in Two-party Split Learning https://arxiv.org/pdf/2102.08504.pdf
> >
> > [2] Differentially Private Label Protection in Split Learning https://arxiv.org/pdf/2203.02073.pdf
> >
> > [3] Deep Learning with Label Differential Privacy https://arxiv.org/pdf/2102.06062.pdf
> >
> > [4] Antipodes of Label Differential Privacy: PATE and ALIBI https://arxiv.org/pdf/2106.03408.pdf

---

> > > ### Comment · Reviewer_7KRM · 2023-11-23
> > > **Thanks for the response**
> > >
> > > I would appreciate the author's rebuttal, which will help improve the overall quality of this paper. Most of my concerns are addressed.
> > >
> > > Regarding the zero-order optimisation, I meant Eq. 2 which recovers the gradient of $g_h$ should be an application of ZOO. Please correct me if I am wrong. So far, I will maintain my score because I think this paper should be further improved, especially from the privacy aspect.

---

### Official Review · Reviewer_aRLa · 2023-10-31

**Soundness:** 1 poor
**Presentation:** 1 poor
**Contribution:** 1 poor
**Rating:** 1
**Confidence:** 4

**Summary:**

This paper focuses on the problem of preserving the privacy of client’s training labels while using fine-tuning APIs.  This paper proposes a fine-tuning protocol that performs Low-Rank Adaptation (i.e., a parameter-efficient fine-tuning) in a setting where clients hold private labels and aim to finetune a model owned by a server without disclosing the labels of examples. The server provides forward and backward passes on their model. The proposed method and its description are very confusing, please see my understanding and comments in the weakness box.

**Strengths:**

The problem of preserving privacy while using fine-tuning APIs is an important problem particularly for large language models given that 1) many recent models are not released, but instead made available as proprietary services; 2) the local resources of clients are limited for fine-tuning.

**Weaknesses:**

My main concern is that the description of the proposed method is confusing and missing lots of information. Figure 2 (which is supposed to be a visualisation of the proposed framework) makes it even more confusing by introducing new variables that were never used in the description. I have spent some time trying to understand and guess the missing information. See below my understanding of the proposed method:
1) a client has local adapters and initializes them locally. How this initialization is done? I can think of two scenarios: 1) the initialization is done randomly; or 2) the initialization is done by copying the weights of adapters owned by the server. Scenario 2 does not make sense because this paper discusses that servers do not want to send their model to clients. Scenario 1 does not make sense either as in step 5 clients use the gradients w.r.t. the server adapter parameters.
2) a client calls forward API call to compute features on each mini-batch of their data. It is not clear how these features are computed. I can think of three different scenarios: 1) the server has both pre-trained model and adapters so the server computes these features as the summation of the output of both of these modules' 2) the server uses only the pre-trained model to compute these features; or 3) the server uses only the adapters to compute these features.
2) a client passes these features to the local “head” and computes task-specific loss function. What is this task-specific loss function?
3) a client computes gradients of the task-specific loss function w.r.t. local head inputs
4) a client passes those gradients to a server via backward API call to compute gradients w.r.t. adapter parameters.
5) a client updates both local adapter parameters and local head parameters. How and which adapters parameters are updated? Please see my points in step 1.

Apart from the above main concern, I have other concerns:

1- Overclaims:
1) This paper claims "privacy guarantees" by saying that "designing a two-party fine-tuning protocol that performs standard parameter-efficient fine-tuning with privacy guarantees". However, there are no privacy guarantees provided, the privacy promise of this paper is ad-hoc and it is just based on increasing the number of servers, assuming they do not collude but assuming that they have the same model.
2) The title of this paper "PRIVACY-PRESERVING LLM FINE-TUNING OVER API" is too generic, oversell and does not represent this work that only considers the privacy of labels.
3) Where "lower performance overhead" is demonstrated "This paper proposes P3EFT, a two-party split learning algorithm that takes advantage of existing PEFT properties to maintain privacy at a lower performance overhead".

2- The observation listed as one of the main contributions at the end of the introduction section ("We observe that, despite fine-tuning less than 0.1% of model parameters, modern PEFT algorithms leak client’s training labels against simple attacks that work for modern pretrained transformers")  and its corresponding Figure 1, has been already demonstrated in existing works such as  Li et al. (2022) even in a more generic way as opposed to simple binary classification tasks that considered in this submission.

3- Not self-contained. For example, a clear description of LoRA which is the main building block of the proposed framework is missing.

4- Not clear what would be the novelty of the proposed privacy-preserving backpropagation in comparison to secret sharing in 2 party computation that have been heavily studied in the literature.

**Questions:**

I have posted many questions regarding the proposed framework, please see the weakness box.

---

> ### Author Response · Authors · 2023-11-22
>
> Thank you for a detailed review!  In the following response, we aim to address your concerns.  Overall, we believe that most of the weaknesses stated in the review are questions and we do our best to answer them below.
>
> > Description of the proposed method
>
> The reviewer asks how LoRA adapters are initialized and how they are used in computations. Overall, we define the multi-party fine-tuning procedure in Section 3.1, which follows standard LoRA fine-tuning. Our algorithm is an extension of this procedure.
>
> Before the training, adapter weights and head weights are randomly initialized on the client side.
>
> During the training on each step:
> 1. adapters first transferred along with the inputs from client to server
> 2. server runs forward pass with given adapters
> 3. server computes activations from the given inputs and adapters and returns them to the client
> 4. client computes loss and **real** gradients w.r.t. to the received activations
> 5. client transfers **noisy** gradients (produced with method described in S3.2) to the server
> 6. server runs backward passes and obtains **noisy** gradients w.r.t to adapter weights
> 7. server transfers **noisy** gradients w.r.t. to adapter weights to the client
> 8. client computes **real** gradients w.r.t. adapter weights
> 9. client executes a step of an arbitrary optimization algorithm.
>
> Overall, we agree that the paper can be improved by adding a dedicated description of the algorithm, and we will improve it in the nearest revision.
>
>
> > What is this task-specific loss function?
>
> For SST2 and MRPC tasks use categorical cross-entropy. In general, this is the loss function used in the standard fine-tuning procedure for downstream tasks, as defined in the supplementary code of the GLUE benchmark paper.
>
> > How and which adapters parameters are updated?
>
> Client updates all adapter parameters after each backpropagation step, which is the standard way of training LoRA adapters defined in Hu et al., (2021) [1]
>
> > There are no privacy guarantees provided, the privacy promise of this paper is ad-hoc and it is just based on increasing the number of servers
>
> While we discuss the privacy of two-party fine-tuning scenario at the end of Section 3.2, we agree that the paper would benefit from making the claims more specific. We will rewrite them in the updated version.
>
> > assuming they do not collude but assuming that they have the same model
>
> We would like to clarify that the second assumption (same model) is relatively common.
> In NLP, there are popular repositories of public models that are used for fine-tuning, e.g. [2] [3]. Most fine-tuning runs reported in NLP research, including most of the runs in the LoRA papers (see [1, 4]) use open-access models. When fine-tuning in this setup, every server can download openly available model weights prior to training, and the client benefits from the shared compute power of the servers. Thus, if the first assumption holds (there are non-colluding servers), then these servers can get the same model.
>
> > The observation listed as one of the main contributions … has been already demonstrated in existing works such as Li et al. (2022) even in a more generic way as opposed to simple binary classification tasks that are considered in this submission.
>
> To the best of our knowledge, Li et al. (2022) also consider the problem of binary classification. Additionally, in the work of Li et al. (2022), the goal is to prevent label leakage through gradients, while our work also considers activations. Finally, we state our contribution as analyzing the privacy of PEFT algorithms (specifically) in practical fine-tuning setups, while Li et al (2022) studies full model fine-tuning. We agree that Li et al (2022) is a highly related work and we cite it as such. However, it does not overlap with our contributions.
>
> > a clear description of LoRA which is the main building block of the proposed framework is missing.
>
> We agree and will include that description in the nearest revision.
>
> > Not clear what would be the novelty of the proposed privacy-preserving backpropagation in comparison to secret sharing in 2 party computation
>
> We appreciate the reviewer's input regarding the technique of secret sharing and agree that it bears some similarities to our method of privacy-preserving backpropagation. However, we maintain that there are important distinctions between us and secret sharing from multi-party  computation. Since another reviewer asked for the same comparison, we will discuss this in more detail in the upcoming General Response.
>
>
>
> [1] https://arxiv.org/abs/2305.14314
>
> [2] https://huggingface.co/
>
> [3] https://pytorch.org/hub/
>
> [4] https://arxiv.org/abs/2305.14314

---

> > ### Comment · Reviewer_aRLa · 2023-11-23
> > **Thank you**
> >
> > I acknowledge that I read your responses.

---

### Author Response · Authors · 2023-11-22

We would like to express our gratitude to all the reviewers for their feedback. We have taken note of your observations and will address all the shortcomings in the next version of the paper.

Based on the insights and comments shared by reviewers, we have come to understand that regrettably, our method's description was overly complex. Therefore, we have decided to clearly separate the thoughts explaining the idea of the method from the final version and present it as a separate algorithm. The revised version includes a more detailed description of forward pass and backpropagation algorithms, as well as several other clarifications as requested by reviewers. For convenience, all modified text is colored blue.

In addition, several reviewers pointed out the similarity between our method for backpropagation and secret-sharing scheme from multi-party computation.
Our technique is indeed related to secret-sharing and we thank reviewers for pointing that out. However, secret sharing is a general cryptographic technique for encrypting information, not for processing it. While there are known extensions [1, 2] that combine secret sharing with e.g. homomorphic encryption for neural network computations, these extensions typically increase computation complexity by at least 2 orders of magnitude (e.g. [1] page 7).

The slowdown is caused by the fact that these techniques have to use MPC-friendly operations instead of the regular (and highly optimized) neural networks layers.
In turn, our approach runs regular (optimized) backpropagation by each party, which is significantly more efficient. However, unlike [1, 2] our technique only works at the backpropagation stage and is useless, e.g. for inference.

Finally, we would like to highlight the main changes we made to the revised version of the paper:

* Backward pass algorithm on page 6

* Full P$^3$EFT fine-tuning algorithm (Appendix A)

* Clear description of LoRA (Appendix B) proposed by reviewer **aRLa**

* Discussion on relevant work Offsite-Tuning: Transfer Learning without Full Model [3] (Appendix C) offered by reviewer **7KRM**

* We have fixed the typos and made clarifications suggested by reviewer **7KRM**

* We have corrected inconsistent statements about the two-party protocol noted by the reviewer **DV4T**

* We have added graphs (Appendix, Figure 6, last row) showing the results of the attack using clustering algorithm (2 cluster k-Means) that interested the reviewer **awQz**.


[1] https://arxiv.org/abs/2109.00984

[2] https://arxiv.org/abs/2110.15440

[3] https://arxiv.org/pdf/2302.04870.pdf

---

### Meta-Review · Area_Chair_4yDK · 2023-12-04

**Metareview:**

The paper proposes a heuristic privacy-preserving method for fine-tuning LLMs.

Strengths: important and timely problem.

Weaknesses: most importantly, the paper lacks a clear definition of what is meant by privacy preservation and how this is guaranteed. This would be necessary to consider acceptance.

**Justification For Why Not Higher Score:**

All reviewers recommend rejection. Paper makes claims that the method is "privacy preserving" without clearly defining what protection is being provided and under what assumptions would it be effective.

**Justification For Why Not Lower Score:**

N/A

---

### Decision · Program_Chairs · 2024-01-16

Reject